# Maynard Smith revisited: A multi-agent reinforcement learning approach to the coevolution of signalling behaviour

**Olivia Macmillan-Scott** [1]*, **Mirco Musolesi**[1,2]

**1** AI Centre, Department of Computer Science, University College London, London, United Kingdom,
**2** Department of Computer Science and Engineering, University of Bologna, Bologna, Italy

\* olivia.macmillan-scott.16@ucl.ac.uk

**Data availability statement:** All code used for setup and running of experiments is available

## Abstract

The coevolution of signalling is a complex problem within animal behaviour, and is also central to communication between artificial agents. The Sir Philip Sidney game was designed to model this dyadic interaction from an evolutionary biology perspective, and was formulated to demonstrate the emergence of honest signalling. We use Multi-Agent Reinforcement Learning (MARL) to show that in the majority of cases, the resulting behaviour adopted by agents is not that shown in the original derivation of the model. This paper demonstrates that MARL can be a powerful tool to study evolutionary dynamics and understand the underlying mechanisms of learning over generations; particularly advantageous is the interpretability of this type of approach, as well as the fact that it allows us to study emergent behaviour without the need to constrain the strategy space from the outset. Although it originally set out to exemplify honest signalling, we show that the game provides no incentive for such behaviour. In the majority of cases, the optimal outcome is one that does not require a signal for the resource to be given. This type of interaction is observed within animal behaviour and is sometimes referred to as proactive prosociality. High learning and low discount rates of the reinforcement learning model are shown to be optimal in order to achieve the outcome that maximises both agents' reward, and proximity to the given threshold leads to suboptimal learning.

## Author summary

When is it too costly for animals to signal that they are in need? Signalling is a crucial part of communication in animal behaviour, and it is also central to other types of interactions, such as those involving artificial agents. We study emergent dynamics in the Sir Philip Sidney game, a game designed to show the mechanisms of honest signalling amongst animals. Using multi-agent reinforcement learning (MARL), we replicate generational learning and show that in the majority of scenarios, the optimal outcome is one of proactive prosociality rather than honest signalling: this is an outcome where a

on a GitHub repository at
https://github.com/oliviams/RL_signalling.

**Funding:** The author(s) received no specific funding for this work.

**Competing interests:** The authors have declared that no competing interests exist.

resource is given without the need for a costly signal. Such behaviour is observed within animal behaviour, most notably among primates. Our results also establish the usefulness of reinforcement learning as a tool to study emergent behaviour and dynamics within animal behaviour for instance as shown here to study behavioural changes and learning over generations.

## Introduction

Signalling is an important mechanism of information transfer between organisms [1], and is also a central aspect of communication [2]. The study of signalling within dyadic interactions, both involving conflicting or common interest, has given rise to a number of theories surrounding the emergence of honest signalling [3,4]. Much of the literature has come from the field of animal behaviour, where signals are crucial to interactions such as parent-offspring begging [5], food sharing [6,7] or sexual selection [3,8]. The latter formed the basis of Zahavi's *handicap principle* [3,9], which argues that signals must be costly in order to be honest. The handicap principle has been shown both theoretically [10], and experimentally [11]: using the example of blue jays, honesty persisted with high signal costs, and disappeared with low costs. Robson's *secret handshake* [4] also studied the emergence of reliable signals within evolutionary games. Perhaps the best known game used to study signalling behaviour within biology is the Sir Philip Sidney game [12], which set out to formalise mechanisms of honest signalling between animals.

Interactions involving signalling have also been modelled from a computational perspective. Catteeuw and Manderick [13] used a reinforcement learning approach to study equilibria in Lewis signalling games [14], where they showed how the use of this method can lead to optimal equilibria. Focusing on prosocial human behaviour, others have identified mechanisms like reputation to be central to indirect reciprocity and to cooperation more widely [15]. Reputation creates an incentive to forego short-term gains of not cooperating in order to benefit from a good reputation in the longer-term; this argument has been demonstrated using reinforcement learning [16,17].

Among artificial agents, communication via signals is essential in achieving coordination. Existing mechanisms include the presence of a coordination protocol [18], or the use of a common language [19]. Communication between agents is particularly central in the field of Cooperative AI [20], where *common ground* is required to ensure agents are able to interpret each other's messages [21]. Under situations where there is no pure common interest, the question of dishonest signals and the potential for deception arises [20,22] - this mirrors the debate within the evolutionary biology literature. Open questions remain regarding how to achieve prosociality among artificial agents in social dilemma-type situations [23]. When it comes to scenarios where there is pure conflict of interest, the incentive to communicate disappears [22]. Evolutionary game theory in conjunction with computational approaches has proven fruitful in studying emergent collective behaviour more widely, although there is still uncertainty relating to the mechanisms leading to behaviour akin to altruism [24].

Given that similar questions arise when it comes to signalling and communication within both artificial agents and animal behaviour, insights derived using computational methods such as reinforcement learning can effectively be applied to understand mechanisms within evolutionary biology. Frankenhuis, Panchanathan and Barto [25] highlight the valuable insights that can be drawn from applying reinforcement learning methods to behavioural ecology, both across generations and within organisms, for instance to study adaptive

behaviour based on experience. Watson et al. [26] also discuss the equivalence of evolutionary processes and simple learning processes, highlighting the use of past experience to make decisions in both types of systems. The value of interactions between the disciplines of evolutionary game theory and machine learning to gain new insights about evolutionary dynamics of interacting agents has previously been discussed [27], in particular with regards to multi-agent systems [28]. Modelling the interaction between two or more agents allows us to study not only behaviour but also emergent strategies, providing us a deeper understanding of decision-making. When reinforcement learning is applied to scenarios involving two or more interacting agents, it is referred to as Multi-Agent Reinforcement Learning (MARL). This is the approach used in this paper, as we will show that it can be a powerful tool for modelling interactions among learning agents. As opposed to other methods, there is no need to hand-code the set of strategies that can be learned by the agent, so we are able to observe the emergence of potentially unanticipated strategies. Reinforcement learning involves a defined action space, which are the possible actions that each agent may take, and a set of rewards that guide the learning process, but no predefined set of strategies. Using this approach, we are able to observe the strategies that emerge through interactions.

An alternative approach is the use of replicator dynamics, a useful method for studying the evolution of strategies within a population. There is considerable work in this area; previous studies have precisely looked at signalling games using replicator dynamics [1,29], in particular at the Sir Philip Sidney game [30]. In their work, Huttegger and Zollman [30] highlight limitations in the traditional evolutionarily stable states (ESS) methodology, and point to the existence of a polymorphic equilibrium involving mixed strategies which, under certain parameters, is more likely to occur than the honest signalling equilibrium. In their work, they also discuss the exclusion in past literature of the strategy 'donate only if no signal' (the emergent strategy we discuss in this paper), and that they believe this to be a mistake. An important limitation is that studying the evolution of behaviour using replicator dynamics requires the use of predetermined strategies and does not allow for strategy innovation. As mutations are not generally incorporated into the model, no new strategies can emerge. Some have addressed this via the replicator–mutator equation [31–33], although this still involves switching from one strategy to another within a predefined set, so the strategy space is constrained and chosen from the outset.

Whereas replicator dynamics is generally concerned with infinite populations, some analyses of the Sir Philip Sidney game on finite populations have also studied the evolutionary stability of honest signalling [34,35]. Catteeuw, Manderick and Han [34] find a lower prevalence of honest signalling in finite populations. They observe that honest signalling is less evolutionarily stable in smaller populations. In a later analysis, they discuss the role of punishment as a behaviour that can foment honest signalling [35]. However, as with replicator dynamics, these approaches also study a set of strategies that is defined and constrained from the outset. The goals of MARL approaches and evolutionary computation models differ, and as such can be complementary. Methods like replicator dynamics allows us to see the relative success and evolutionary stability of given strategies at population level, whereas reinforcement learning looks at individual agents and studies the optimal policy that they learn through repeated interactions. In the latter, agents can dynamically update their strategy in response to the environment, as animals are able to adapt to shifting conditions. Replicator dynamics is a useful method to understand the behaviour of populations over time, whereas reinforcement learning more closely mimics individual learning. Another distinction is that, whereas replicator dynamics aims to study which strategies are selected from a fixed and predefined set, reinforcement learning instead allows us to understand which strategies emerge from a (much larger) strategy space delimited by the agent's architectural constraints.

This is where reinforcement learning presents an advantage for studying behavioural dynamics. In using MARL, there is no requirement to constrain the set of possible strategies that agents may learn beyond architectural constraints imposed by the agent's design, primarily memory length. In this way, we can observe the emergence and coevolution of new strategies in a population, rather than the dynamics of static strategies. Some have even proven that replicator dynamics can emerge from reinforcement learning algorithms [36–39]. Reinforcement learning has been discussed as a useful tool to study animal behaviour and to model their learning through trial and error [25,40–42]. In the multi-agent setting, it has been used to study animal behaviour in areas including predatory-prey dynamics [43,44] and collective/swarm movement [45,46].

We precisely set out to study the mechanisms of signalling in the Sir Philip Sidney game through MARL; this approach allows us to observe the coevolution of behaviour, identify emergent strategies, and explore how the outcomes change depending on the variation of certain parameters. The use of reinforcement learning, more specifically Q-learning in this case, only requires a defined action space, environment, and a set of learning parameters. By not requiring the strategy space to be defined from the outset, we can observe the strategies that emerge through repeated interactions. Agents initially explore randomly between the set of actions available to them, and update the expected reward of a given action. In this way, reinforcement learning allows the agents to arrive at the optimal or near-optimal strategy. We thus show that MARL can be a powerful tool to study evolutionary dynamics and understand the underlying mechanisms of learning over generations. In contrast to other methods, reinforcement learning does not require the strategy space to be constrained beyond the set of learnable strategies defined by the agent's architecture, so is a powerful tool to understand the emergence and learning of decision-making strategies - in this case, with the presence of signalling. This approach can be widely generalised to study the inter-relationships between signalling and the emergence of strategies in a population.

In this paper, we use reinforcement learning to study the Sir Philip Sidney game, a game in which Maynard Smith set out to demonstrate the dynamics of honest signalling [12]. We show that when the parameters of the model are within a given threshold, agents learn to arrive at the optimal equilibrium where the resource is given without the need for a signal; this is akin to Huttegger and Zollman's [30] aforementioned 'donate only if no signal' strategy. In animal behaviour, this can be assimilated to proactive prosociality, and is observed across species. Various mechanisms may lead to this observed behaviour, including kin selection or social learning. As the parameter values approach the threshold, the percentage of agents that learn the optimal strategy significantly decreases, although it is still learned in the majority of cases. High learning and low discount rates result in the agents learning the optimal strategies in the highest proportion of runs.

## Methods

### The Sir Philip Sidney game

The Sir Philip Sidney game was developed by Maynard Smith as a way to model signalling within animal behaviour [12] - in particular, the aim of the model was to illustrate the mechanisms of honest signalling initially proposed by Zahavi [3,9] and subsequently developed by others [10,47]. This framework has been the subject of several studies, particularly within the biological sciences [6,48–50]. Whereas most models assume infinite populations; it is worth noting that an analysis of the game in finite populations found much lower prevalence of honest signalling [34]. The game involves two agents/players: a *signaller* and a *receiver*. In the original formulation, the players are referred to as the Beneficiary (*B*) and Donor (*D*)

respectively. *B* represents an agent who may or may not be in need of an indivisible resource, whereas *D* holds this resource. Taking this resource to be water, *B* may be in one of two states: thirsty (with probability *p*) or not thirsty (with probability 1–*p*). *B* knows the state they are in, while *D* can observe *B*'s actions but not state. Each player has two possible actions: *B* may signal or not signal that they are in need of the resource, and in response *D* may give or keep said resource. The set of actions $\mathcal{A}$ for each player can therefore be represented as $\mathcal{A}_B = \{S, N\}$ and $\mathcal{A}_D = \{G, K\}$. If *D* keeps the resource, their chance of survival is 1; this decreases to *U* if they give the resource, where $0 < U < 1$. For *B*, the chance of survival is as follows: 1 if they get water, regardless of their state, 0 if they are in a thirsty state and do not get water, and *V* if they are in a not thirsty state and do not get water, where $0 < V < 1$.

As the game was designed to model animal behaviour, and therefore simulates learning over generations, we consider inclusive fitness rather than individual payoff; the use of inclusive fitness instead of individual reward is in line with Maynard Smith's original formulation [12] and subsequent studies [6,30,34]. Given the relatedness coefficient (*r*) between both players, the inclusive fitness (*F*) of Player *i* is calculated as the sum of a player's individual payoff $P_i$ and the opponent *j*'s payoff $P_j$ multiplied by a factor of *r*: $F_i = P_i + rP_j$. The extensive-form game tree is presented in Fig 1, where a dashed line represents an information set. Nodes within the same information set (filled in the same colour for clarity) represent those where Player *D* does not know which of the two they are at, as they do not observe *B*'s state. Since its application to animal behaviour, the Sir Philip Sidney model has been used to study other dyadic interactions that involve a signal between the two agents; the game has also been expanded beyond the use of discrete parameters, to a continuous form [6]. In this article, we will also use a continuous version of the game to illustrate threshold values for each outcome.

In the original formulation of the game, Maynard Smith derived a set of evolutionarily stable strategies [12]. For the signaller, he argued it is evolutionarily stable to signal honestly (only signal if they are indeed in need of the resource), and for the receiver to give the resource only if the other player signals. A set of inequalities were presented for the different parameters, stating that if these are met, the aforementioned equilibrium will hold. However, it has been shown both theoretically [48] and through simulation [49] that an outcome was overlooked: there exists an equilibrium where one agent never signals, and the other always gives the resource regardless. Hamblin and Hurd [49] refer to this equilibrium in their

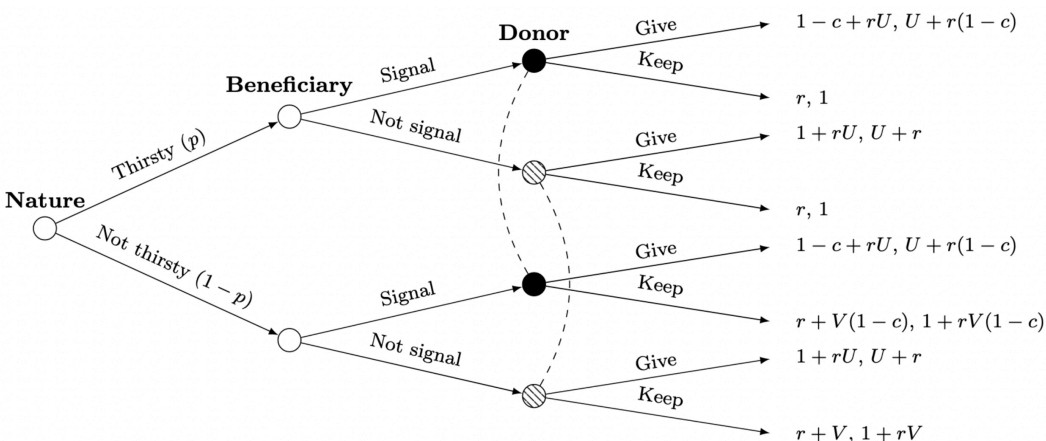

**Fig 1. Extensive-form game tree: Sir Philip Sidney game.**

work as the 'non-communicating, always-give' set, reached by Huttegger and Zollman's [30] aforementioned 'donate only if no signal' strategy. This outcome would be reached if both players try to maximise their inclusive fitness - payoffs in this equilibrium are higher than if pursuing a strategy of dishonest signalling [49]. A full mathematical derivation of this solution is included in S1 Appendix. As Huttegger and Zollman [30] point out, this equilibrium has been intentionally excluded from previous literature [48,51], but doing so fails to consider emerging dynamics under certain parameters.

In some cases, it may be valid to exclude the possibility of the donor giving the resource without the beneficiary signalling that they are in need, and so excluding the equilibrium discussed in this paper. These are cases where the resource being given is conditional on a signal, examples of which can be found in animal behaviour. In this case, we have chosen not to assume conditionality, as there are also examples in animal behaviour of a resource being given without the need for a signal. One example of such a strategy combination is found in Rodríguez-Gironés et al.'s [52] model of begging between offspring and parents: they find an equilibrium where chicks do not beg for food, and parents still provide optimally. This type of interaction is sometimes referred to in the literature as proactive prosociality, and is also observed among primates in the form of allomaternal care, as well as in the behaviour of other animal groups [53,54]. By not assuming conditionality on the beneficiary's signal, we are also including the possible action combination of {*Not Signal, Give*} that has been excluded in previous literature. This is not to say that the donor's action is no longer conditional on the beneficiary's. The game remains sequential, and the donor still observes the beneficiary's action before taking their own. The only difference is that, whereas following previous literature the donor would not have been able to give the resource in the absence of the signal, we do not impose this condition.

This paper sets out to show that the expected evolutionarily stable strategies of the Sir Philip Sidney game would be those where the resource is given without the need for a signal; these results have previously been discussed theoretically by Bergstrom and Lachmann [48]. The methodology will be using reinforcement learning agents playing the game against each other, and observing the resulting strategies. We will show that learning agents playing the Sir Philip Sidney game do not learn the strategies deemed evolutionarily stable by Maynard Smith; instead, as we expected, they learn to either not signal and give regardless, or not signal and keep the resource (depending on the parameter values). Hamblin and Hurd [49] show a similar result through the use of a genetic algorithm. As with replicator dynamics, their approach requires a set of predetermined strategies. By using reinforcement learning, we do not constrain the strategy space and can instead observe the strategies that emerge from the agents' own learning of optimal actions. Constraints on possible learned strategies are only imposed by the agent's architecture, such as memory length.

The motivation behind the use of reinforcement learning is in the explanatory value of this type of approach, as well as the fact that this approach does not require the use of predetermined or hand-coded strategies. Referring to the Philip Sidney game, Johnstone and Grafen stated that "signal and response behaviour therefore co-evolve, and the outcome of this coevolution is difficult to predict" [6, p. 215]. With the use of reinforcement learning, we can observe this coevolution and explore how the outcomes change depending on the variation of certain parameters, allowing us to precisely study this behaviour that the authors refer to as difficult to predict. In this way, the learning agents simulate learning over generations and evolution: by applying reinforcement learning to animal behaviour, we can analyse the behaviour of generations of animals that are related to each other, and look at how much the ensuing behaviours are affected by and dependent on the behaviour of others.

To ensure that the findings are robust to varying parameters, for each set of values we show both the case where $p = 0$ and $p = 1$, representing the agent in need of the resource with probability 0 and 1 respectively. The value of $p$ does not affect the results - this was demonstrated by Maynard Smith in the original formulation of the model [12], and we find that this holds for reinforcement learning agents.

## Revised equilibria of the Philip Sidney game

As we have seen, the Philip Sidney game was originally intended to demonstrate honest signalling; provided the parameter values met certain conditions, it was argued that the evolutionarily stable strategies were for the Beneficiary to signal when they are in need (honest signalling), and for the Donor to give the resource when the Beneficiary signals. However, the original derivation overlooked a crucial outcome. For a majority of possible parameter values, the evolutionarily stable strategies would be for the Beneficiary not to signal, and the Donor to share the resource regardless [48]. When both players are choosing the actions that would maximise the inclusive fitness, this is the resulting combination of strategies. Within animal behaviour, this can be equated to proactive prosociality, often observed during allomaternal care in primates [53,55]. This observed behaviour may be caused by differing underlying mechanisms, such as kin selection or social learning.

The alternative set of evolutionarily stable strategies can be derived from the inclusive fitness matrices for each of the two possible states. As mentioned, $B$ may be in one of two states: *thirsty* with probability $p$ (Table 1), or *not thirsty* with probability $(1-p)$ (Table 2).

From Tables 1 and 2 we can compare $B$'s payoffs if they choose to *Signal* or *Not signal* when keeping $D$'s actions constant (so comparing $B$'s payoffs in the top and bottom rows of each matrix). It is clear that there is no situation where $B$ would prefer to signal, provided that $c \leq 0$ and $V \leq 0$, which must be satisfied: for instance, for $p = 1$ (Table 1), it must always be true that $1+rU > 1-c+rU$. Given that $B$ never signals, $D$ would choose to *Give* if the following condition is met (see S1 Appendix):

$$r > \frac{1 - U}{1 - V + pV} \tag{1}$$

In order to analyse the emergent behaviour when the above threshold is met or not, we use various combinations of parameters in our experiments. The values were chosen to exemplify three different cases: the first where Eq 1 is not satisfied, so we do not expect a cooperative outcome as other strategies of $D$ will invade, and the second and third show two cases where the inequalities are satisfied. However, the third case ($U = 0.95, V = 0.75, r = 0.9$) exemplifies a

**Table 1. Matrix of inclusive fitness given $p = 1$.** $B$ is the row player and $D$ is the column player.

|            | Keep | Give |
|------------|------|------|
| Signal     | $r, 1$ | $1 - c + rU, U + r(1 - c)$ |
| Not signal | $r, 1$ | $1 + rU, U + r$ |

**Table 2. Matrix of inclusive fitness given $p = 0$.** $B$ is the row player and $D$ is the column player.

|            | Keep | Give |
|------------|------|------|
| Signal     | $r + V(1 - c), 1 + rV(1 - c)$ | $1 - c + rU, U + r(1 - c)$ |
| Not signal | $r + V, 1 + rV$ | $1 + rU, U + r$ |

scenario where the values lie close to the limits. We included this to explore how proximity to the threshold affects the learned strategies of Player $D$ in particular - as the parameter values lie close to the limit, we expect to see the optimal strategy being learned a lower proportion of the time. This is because the difference in inclusive fitness from playing $G$ or $K$ will be smaller in absolute terms, therefore there will be less difference between the learned Q-values. It is worth emphasising that throughout this analysis we are considering inclusive fitness; were we to look at individual payoff, there would be no scenario where it is evolutionarily stable for $D$ to give the resource.

Table 3 illustrates the parameter values that we will use in our simulation, whether or not they satisfy inequality Eq 1, and the strategies that we expect the reinforcement learning agents to learn (assuming each player is maximising their inclusive fitness). For the example where $U = 0.9$, $V = 0.1$ and $p = 0.5$, Fig 2 shows the parameter values in relation to the threshold inequality presented above. From the graph, we can see that $r = 0.8$ lies within the shaded area with both low and high signal costs, whereas for high signal costs ($c = 0.75$), lower values of $r$ would no longer satisfy this condition.

**Table 3**. **Alternative categorisation of example parameter values that satisfy each set of inequalities.** For expectation, $\{N, K\}$ means the Beneficiary does not signal, and the Donor keeps the resource. Valid for any value of $p$.

| $r$ | $V$ | $U$ | *Eq 1 satisfied* | *Expectation* |
|-----|-----|-----|------------------|---------------|
| 0.5 | 0.2 | 0.2 | No | $\{N, K\}$ |
| 0.8 | 0.1 | 0.9 | Yes | $\{N, G\}$ |
| 0.9 | 0.75 | 0.95 | Yes | $\{N, G\}$ |

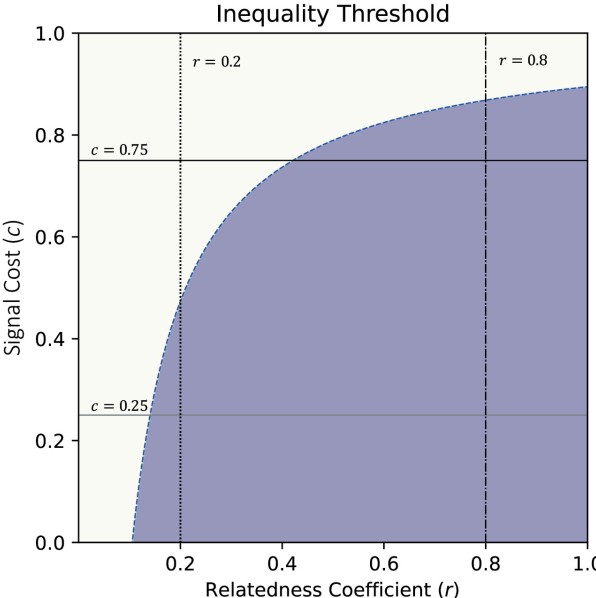

**Fig 2. Visual representation of threshold values where the evolutionarily stable strategies presented in this article would hold** ($p = 0.5$, $U = 0.9$, $V = 0.1$). The intersection of $r = 0.8$ with both $c = 0.25$ and $c = 0.75$ is within the shaded area of the two inequalities, therefore both thresholds are satisfied and we would expect the resulting strategy combination to be $\{N, G\}$. However, for $r = 0.2$, the evolutionarily stable strategies of *Not signal* and *Keep* only hold for low signal costs as the intersection with $c = 0.75$ is outside the shaded area.

As shown above, there are no set of values that would result in *Signal* being the preferred strategy for the Beneficiary. Even as we vary signal costs, as long as $c>0$ is satisfied, the beneficiary would still choose not to signal that they are in need. Therefore, it is only the behaviour of the Donor that we expect to see changes in, illustrated in Table 3. Depending on the parameter values, it may be evolutionarily stable or not for the Donor to give the resource - there are cases, particularly where $r$ is high enough, that inclusive fitness is maximised by giving the resource (even though individual payoff is not).

### Coevolution of reinforcement learning strategies

We will use MARL to demonstrate that the above equilibria are reached by learning agents. In order to more closely approximate the model to a continuous signalling game [6], and to ensure that results found are robust to increased signal strengths, we include two possible signal costs. This means that the Beneficiary can either send a strong or weak signal. The Beneficiary player class therefore has three possible actions: *Not signal, Weak signal, Strong signal*, corresponding to the different signal costs detailed above ($c = 0.25$ for a weak signal and $c = 0.75$ for a strong signal). The reason for including weak and strong signals is to closer approximate the model to a continuous signalling game [6], and to ensure that results found are robust to increased signal strengths. The Donor player class, as in Maynard Smith's original model, has two possible actions: *Give, Keep*.

In the simulation, the agents update their expected value of an action through the use of the Q-learning algorithm [56]. At time $t$, each player has an action $a$ and state $s$ - this state is made up of both players' actions in the previous round. $R_t(s, a)$ represents the reward for a given state-action combination - in our case, this is the inclusive fitness. $\alpha$ is the learning rate and $\gamma$ is the discount rate, both of which lie between 0 and 1; the choice of values for both of these parameters will be discussed below. To assess the action to be taken at each stage, the following Q-learning formula is used:

$$Q_t(s, a) = Q_{t-1}(s, a) + \alpha(R_t(s, a) + \gamma \max_{a'} Q_{t-1}(s', a') - Q_{t-1}(s, a)) \qquad (2)$$

where $s'$ and $a'$ represent the new state and action at time $t + 1$. Each action-state combination is therefore assigned a value ($Q_t(s, a)$, referred to as a Q-value); these Q-values are then used to make decisions about which action should be taken next. The agent updates the Q-value $Q_t$ for the action they just played once they know the reward they received, meaning that they know the action just played by the opponent and so the state they will be in when taking the next action. The term that is multiplied by the discount factor $\gamma$ represents the value of the action with maximum expected reward given the state they will be in when taking this next action.

Before each interaction, both agents' Q-value tables are initialised at 0. Each dyad plays 500 rounds of the game, as this allows for the Q-values to converge, so we can clearly show what the learned strategies are. The first 50 rounds of each interaction between two agents are exploratory ($\varepsilon = 1$), meaning that both learning agents are playing randomly while updating their Q-value tables. After these 50 rounds, the rate of exploration decays at a rate of $\varepsilon = \frac{2}{N}$, where $N$ is the number of rounds played. For each parameter set, we run 1000 iterations, and calculate the average values over all iterations. The Philip Sidney game was designed to model the coevolution of signaller-receiver behaviour over generations within animal behaviour. Using a reinforcement learning approach, each round of the game can be understood as a generation in the animal species; however, reinforcement learning is also used to model individual animal behaviour, so the simulation could also be understood in this way.

We also explore the effects of varying the learning and discount rates on the resulting behaviour of the agents. We used three values for the learning rate: 0.1, 0.5 and 0.9, and each of these was run with two different discount rates: 0.1 and 0.9. (for a greater range of values for both learning and discount rates, see S2 Appendix). A higher learning rate accelerates the speed of learning, as Q-values are updated at a faster rate, but may result in convergence at a suboptimal value. On the other hand, a high discount factor gives more weight to longer-term rewards, therefore we expect that the Q-values of agents with a low discount rate will converge faster.

## Results

### Learned strategies of RL agents

For each set of parameter values, the strategy most often learned by the reinforcement learning agents under optimal learning and discount rates is precisely the strategy that we expected through the derivation of evolutionarily stable strategies (see Table 4). When inequality *(1)* is not met, the resulting behaviour displayed by the agents is $\{N, K\}$: the Beneficiary does not signal, and the Donor keeps the resource. We have seen that the behaviour of the Beneficiary is never expected to deviate from *N* (*Not signal*), as there is no incentive to do so; it is only the behaviour of the Donor that changes. In contrast, when *(1)* is satisfied, the resulting strategy set is $\{N, G\}$: the Beneficiary again does not signal, but the Donor does give the resource regardless of the lack of signal received. These results are robust irrespective of the state of the Beneficiary, i.e. whether they are in a *thirsty* or *not thirsty* state, showing that the value of *p* does not affect the results, as expected by Maynard Smith [12].

Although we show here the case where both agents are reinforcement learning agents with equal parameters, we also wanted to explore the emergent behaviour of a learning agent when interacting with an agent that is following a predefined strategy. As this is not the direct aim of this paper, we have included examples in the Appendix where one of the agents follows a static strategy (see S3 Appendix). In particular, no matter the choice of parameters, when using two reinforcement learning agents we always have a case where Player *B* most often chooses to *Not Signal*. Therefore, by implementing predetermined strategies, we can see what is learned by Player *D* when their opponent either chooses to *Weak Signal* or *Strong Signal* a majority of the time. Additionally, we show how the resulting strategies for both players vary with longer memory of past actions (see S4 Appendix); in this case, we see a greater variation of strategies learned, where the agent alternates or cycles through a pattern of actions.

### Optimal RL parameters

Figs 3 and 4 show the resulting strategies learned by the RL agents under the parameters $S = 0.2$, $V = 0.2$, $r = 0.5$ (Case 1) for Players *B* and *D* respectively; Tables 5 and 6 describe the corresponding strategies. The strategies presented in these tables are those most often learned by the reinforcement learning agents that appear in the corresponding graphs. This is not the

**Table 4. Resulting strategies of learning agents, showing the top strategy learned and the proportion of runs this was the resulting strategy for the parameters LR=0.9, DR=0.1.** For expectation, $\{N, K\}$ means the Beneficiary does not signal, and the Donor keeps the resource. Valid for any value of *p*.

| r | V | U | (1) satisfied | Expectation | B | D |
|---|---|---|---|---|---|---|
| 0.5 | 0.2 | 0.2 | No | $\{N, K\}$ | N = 83.3% | K = 97.1% |
| 0.8 | 0.1 | 0.9 | Yes | $\{N, G\}$ | N = 46.1% | G = 94.6% |
| 0.9 | 0.75 | 0.95 | Yes | $\{N, G\}$ | N = 82.4% | G = 48.2% |

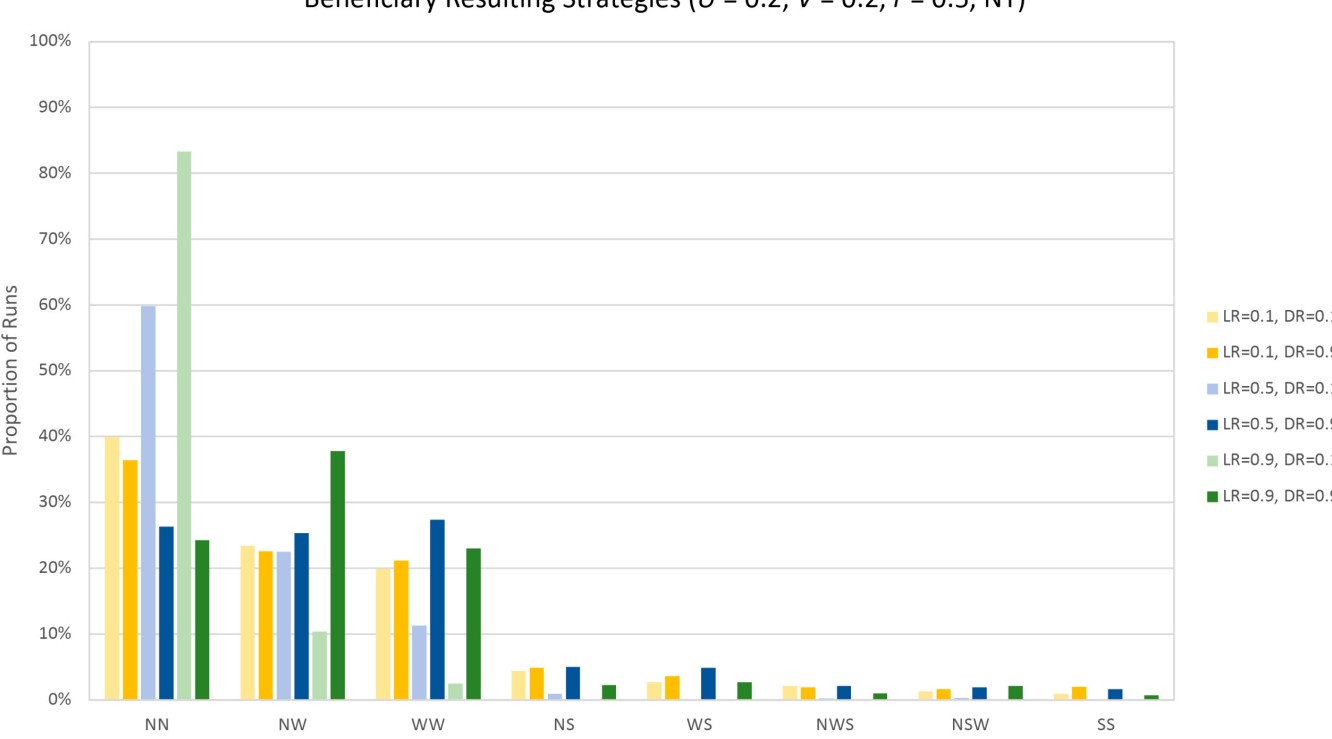

**Fig 3. Beneficiary resulting strategies.** Percentage of runs that the Beneficiary learns each strategy. *Not thirsty* state, $r = 0.5$, $U = 0.2$, $V = 0.2$. Strategies described in Table 5.

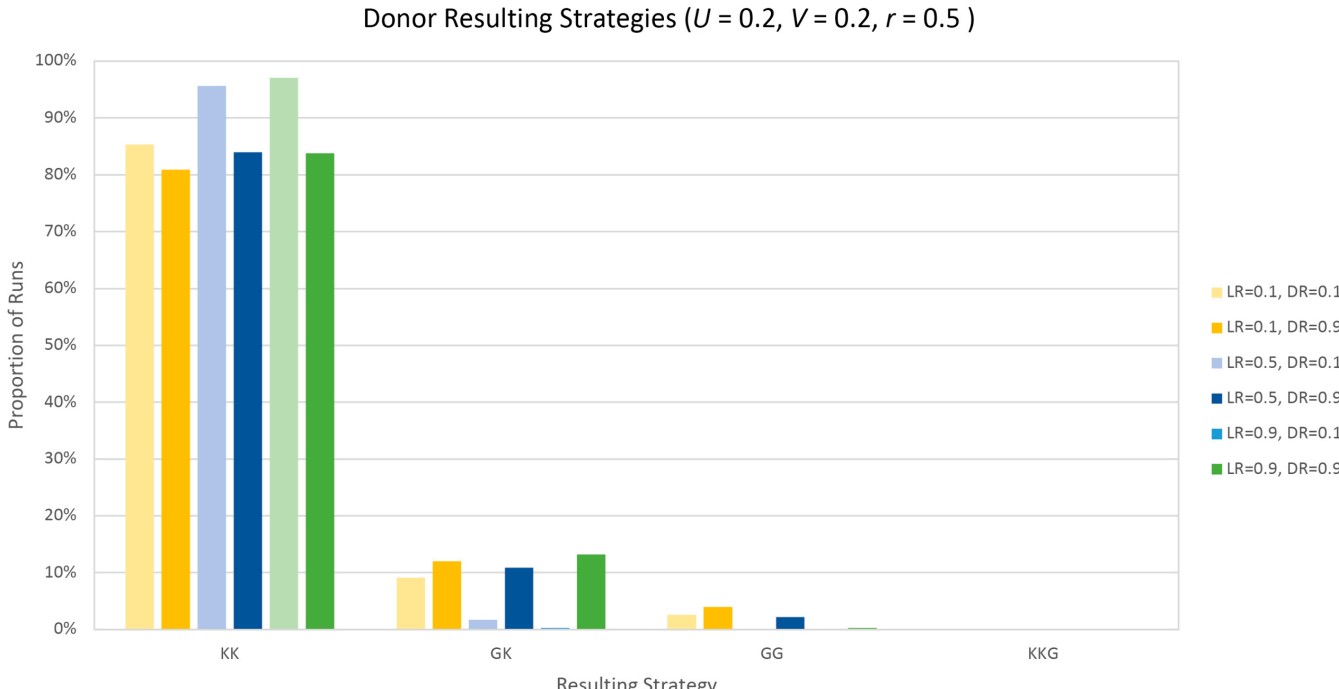

**Fig 4. Donor resulting strategies.** Percentage of runs that the Donor learns each strategy. *Not thirsty* state, $r = 0.5$, $U = 0.2$, $V = 0.2$. Strategies described in Table 6.

**Table 5. Strategies most often learned by the Beneficiary with parameters *S* = 0.2, *V* = 0.2, *r* = 0.5 (Case 1), see Fig 3.**

| Acronym | Strategy |
|---------|----------|
| NN | Never signal |
| NW | Alternate between *Not signal* and *Weak signal* |
| WW | Always signal weakly |
| NS | Alternate between *Not signal* and *Strong signal* |
| WS | Alternate between *Weak signal* and *Strong signal* |
| NWS | Cycle through *Not signal*, *Weak signal* and *Strong signal* |
| NSW | Cycle through *Not signal*, *Strong signal* and *Weak signal* |
| SS | Always signal strongly |

**Table 6. Strategies most often learned by the Donor with parameters *S* = 0.2, *V* = 0.2, *r* = 0.5 (Case 1), see Fig 4.**

| Acronym | Strategy |
|---------|----------|
| KK | Always keep |
| GK | Alternate between *Give* and *Keep* |
| GG | Always give |
| KKG | Cycle through *Keep*, *Keep*, *Give* |

full set of possible strategies, as through MARL, any action combination can be learned. For both agents, a learning rate of 0.9 and discount rate of 0.1 appear to be the optimal combination, as it is with this choice of parameters that the expected strategy is learned most often. From Fig 3, we can see that *B* very rarely learns a strategy involving strong signalling (*S*); this is in line with our previous derivation, as the action *S* reduces *B*'s expected inclusive fitness the most.

For Player *D*, the difference in learned strategies shows less variation with different learning and discount rates (Fig 4). For the values of *S*, *V* and *r* shown, the optimal strategy is to always keep the resource, which is what we observed at least 80.9% of iterations. The combination of parameters resulting in this lowest value (80.9%) is a learning rate of 0.1 and a discount rate of 0.9. For each value of learning rate, a lower discount rate leads to the optimal strategy being learned a higher proportion of the time.

For Player *B*, there is more variation due to there being three possible actions and less difference in the absolute value of the rewards. Nevertheless, we observe the same result: for the same learning rate, a discount rate of 0.1 always outperforms a discount rate of 0.9. As the Philip Sidney game was formulated to model interactions within animal behaviour, we can interpret this result with respect to the original application. A lower discount rate can be assimilated to maximising the importance given to immediate survival rather than potential long-term gains. As in this case we are playing repeated instances of the same game, maximising short-term survival will ultimately also maximise long-term payoffs, therefore agents with a lower discount rate learn the optimal strategies a higher proportion of the time.

## Case comparison

*D* learns the strategy *Always Give* in Case 2 almost twice as often as in Case 3. The reason for this is clear when looking at the graphs of Q-values over the rounds. The shape of the graph with a convergence around round 50 is due to the fact that agents play exploratory actions for the first 50 rounds, after which they play according to the learned Q-values. Figs 5A and 6A illustrate the average Q-values for each action across the rounds for *B*, as well as the average for each state; Figs 5B and 6B show the same for *D*. Whereas for Case 2, the Q-values of

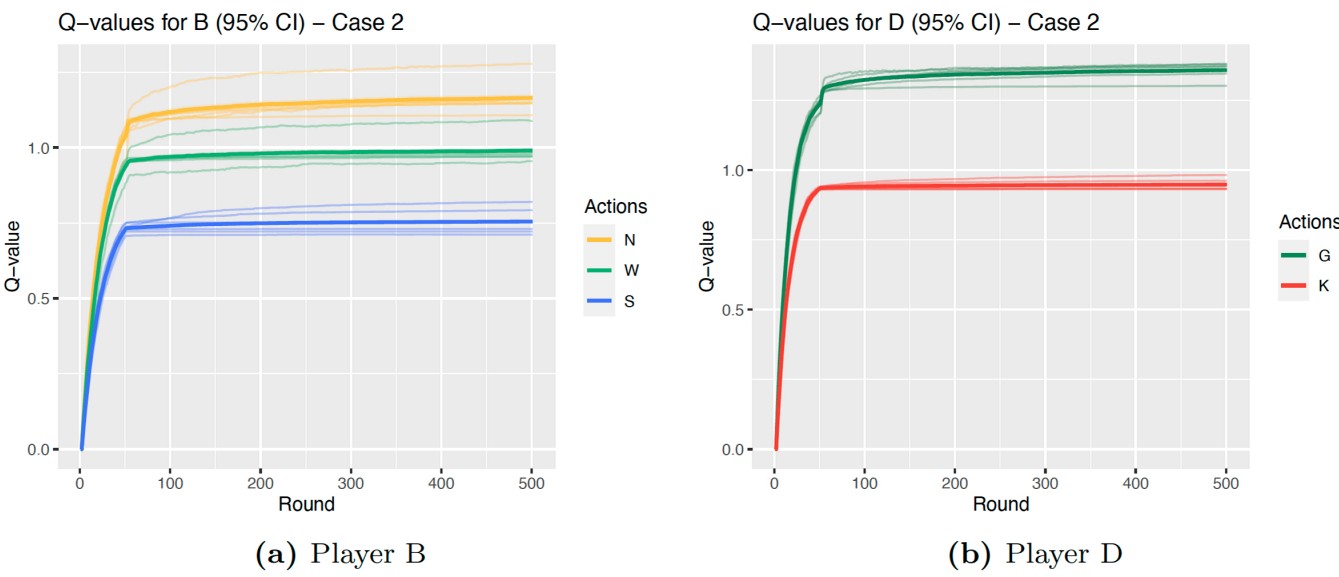

**(a)** Player B

**(b)** Player D

**Fig 5. Q-values Case 2, darker line is average across all, faint lines are average for each state.**

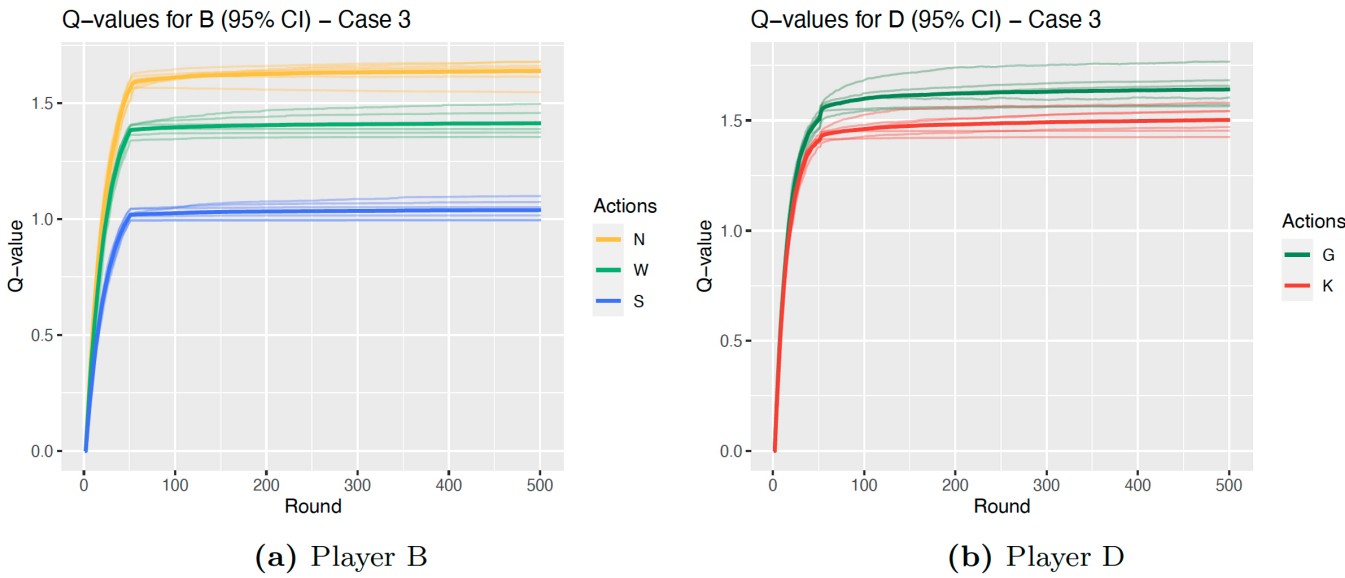

**(a)** Player B

**(b)** Player D

**Fig 6. Q-values Case 3, darker line is average across all, faint lines are average for each state.**

*D* associated with the action *G* quickly become distinctly higher than for *K*, Case 3 portrays a different scenario. While overall the agent does learn that *G* will result in a higher payoff than *K*, the difference is much smaller. Not only is the difference between the averages much less marked (0.410 for Case 2 and 0.139 for Case 3), we can also see that the average Q-value for *K* for one state (*NG*) is higher than that for *G* in two out of the six states (*WG* and *SG*). As mentioned above, this is to be expected as the parameters for Case 3 are much closer to the threshold and parameter limits, therefore the difference in inclusive fitness is smaller than

for Case 2. Nevertheless, the most learned strategy is still to *Always Give*, and the next most learned is to alternate between *G* and *K*, which is learned in 38.5% of iterations.

A similar outcome is observed for Player *B*, where in Case 2 the optimal strategy is only learned in 46.1% of iterations. Again, this is clear in Fig 5A, where the state with the highest average Q-values for *W* (state *NG*) is very close to the lowest for *N* (state *SK*). In particular, the state with the highest Q-values for *W*, *NG*, is observed often by *B* as for this set of parameters Player *D* learns to *Always Give* 94.6% of the time. Therefore, Player *B* will choose action *W* more often than is optimal.

## Discussion

The results presented above are in line with the theoretical derivation of evolutionarily stable strategies (see [48] and S1 Appendix). Sets of parameters that satisfy Eq 1 will result in the strategies $\{N, G\}$ in a majority of cases, whereas those that do not satisfy this inequality result in strategies $\{N, K\}$. This is in accordance with our expectations, as *B* never has an incentive to deviate from the strategy *Never Signal*; it is therefore only *D*'s strategy that varies according to the inequality threshold.

The use of reinforcement learning allows us to observe the coevolution of behaviours and strategies between the two learning agents, which simulates learning over generations and evolution within animal behaviour. Through a theoretical analysis of the Sir Philip Sidney game, we can derive the evolutionarily stable strategies, and predict what the dominant behaviours will be. However, the use of reinforcement learning provides additional complexity as we can also study what other strategies emerge, and under what conditions we are more likely to observe a greater proportion of the population learning suboptimal behaviours. As opposed to replicator dynamics, by using MARL we do not need to hand-code the learnable set of strategies and can observe the emergent behaviour of the learning agents. The only constraints are imposed by the agent's architecture, primarily memory length.

With regards to the optimal combination of learning and discount rates, we show that a lower discount rate always results in the highest inclusive fitness being achieved by the RL agent. The optimal combination for our formulation of the model is a learning rate of 0.9 and discount rate of 0.1. In some cases, the expected strategy is not learned as often as we would expect (although it is always learned more often than any other). We observe this phenomenon in Case 2 for *B* and Case 3 for *D*: this happens more markedly as the parameter values approach the limit given by Eq 1. One contributing factor to this result may be the length of the exploration period. Having either an initial exploration period longer than 50 rounds or slower decay of $\varepsilon$ can allow for a greater distance to be learned between Q-values, and therefore the optimal strategy to be learned a greater proportion of the time. We show examples of outcomes with varying length of initial exploration period and varying $\varepsilon$ decay in S5 Appendix.

## Conclusion

The approach presented in this paper illustrates how MARL can be used to study the coevolution of strategies and behaviours, and, in particular, animal behaviour. Whereas Maynard Smith initially formulated the Sir Philip Sidney game as a way to exemplify honest signalling [12], we show that in a majority of cases there is an alternative outcome. This result has previously been discussed in the literature [48,49,52] and relaxes the assumption of conditionality: the equilibrium shown in this paper is reached when we remove the condition that the resource cannot be given in the absence of a signal. The game remains sequential, with *D* only

acting after observing $B$'s action. The resulting strategy combination where $B$ never signals but $D$ always gives the resource regardless can be assimilated to proactive prosociality. We observe this phenomenon both in the animal realm [53,54], and within human behaviour [55], and this behaviour may be caused by a number of mechanisms.

In all cases, the learning agents arrive at the optimal strategy a majority of the time; regardless of the chosen parameters, the Beneficiary predominantly learns a non-signalling strategy. For the Donor, depending on whether or not the threshold values are exceeded, it will either learn to give the resource without a signal or to keep the resource. The former case is also studied by Hamblin and Hurd [49], who refer to this as the 'non-communicating, always-give' set, and by Huttegger and Zollman [30], as the 'donate only if no signal' strategy. As parameter values approach the limit given, the proportion of runs where the optimal strategy is learned decreases - this is as we would expect, as the difference in resulting inclusive fitness between strategies becomes smaller. High learning and low discount rates result in the highest proportion of agents learning the expected strategies.

Therefore, we show that MARL can be a powerful tool to study evolutionary dynamics and understand the underlying mechanisms of learning over generations without the need to choose the strategy space from the outset (beyond the set of learnable strategies). Methods such as the one presented in this paper simulate learning over generations by incrementally updating the beliefs of artificial agents, and allow us to explore how outcomes change depending on the variation of certain parameters. The application of this type of approach can be an effective way to explore the emergent behaviours within animal populations, in particular looking at how behaviours coevolve. The model parameters vary in how literally they can be interpreted when applied to animal behaviour: whereas learning and discount rates require more interpretation, the relatedness coefficient and signal costs can be directly understood in terms of evolutionary biology,

The above study was intended to derive the equilibrium for the Sir Philip Sidney game through the use of reinforcement learning, and therefore focuses on dyadic interactions. However, this focus leaves a series of open questions. Further extensions to this research could explore population dynamics, perhaps including partner selection (see for example [57]). Another interesting area of research could further explore dynamics of learning over generations, where agents are initialised with the learned values of a 'parent' agent.

## Supporting information

**S1 Appendix. Derivation of evolutionarily stable strategies.**
(PDF)

**S2 Appendix. Results with varying learning and discount rates.**
(PDF)

**S3 Appendix. Results with static agents.**
(PDF)

**S4 Appendix. Results with varying memory.**
(PDF)

**S5 Appendix. Results with varying exploration.**
(PDF)

## Author contributions

**Conceptualization:** Olivia Macmillan-Scott, Mirco Musolesi.

**Formal analysis:** Olivia Macmillan-Scott.

**Methodology:** Olivia Macmillan-Scott.

**Software:** Olivia Macmillan-Scott.

**Supervision:** Mirco Musolesi.

**Visualization:** Olivia Macmillan-Scott.

**Writing – original draft:** Olivia Macmillan-Scott.

**Writing – review & editing:** Mirco Musolesi.

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
