## [Decision Letter · Decision Letter 0]

22 Jan 2025

PCOMPBIOL-D-24-02076

Maynard Smith revisited: A multi-agent reinforcement learning approach to the coevolution of signalling behaviour

PLOS Computational Biology

Dear Dr. Macmillan-Scott,

Thank you for submitting your manuscript to PLOS Computational Biology. After careful consideration, we feel that it has merit but does not fully meet PLOS Computational Biology's publication criteria as it currently stands. Therefore, we invite you to submit a revised version of the manuscript that addresses the points raised during the review process.

Please submit your revised manuscript within 60 days Mar 24 2025 11:59PM. If you will need more time than this to complete your revisions, please reply to this message or contact the journal office at ploscompbiol@plos.org. Please include the following items when submitting your revised manuscript:

We look forward to receiving your revised manuscript.

Kind regards,

Feng Fu

Academic Editor

PLOS Computational Biology

Tobias Bollenbach

Section Editor

PLOS Computational Biology

**Additional Editor Comments :**

While both reviewers see the strengths and contributions of the manuscript, they also raise important points. In your revisions, it would be necessary to address their comments for further consideration.

**Journal Requirements:**

At this stage, the following Authors/Authors require contributions: Olivia Macmillan-Scott, and Mirco Musolesi. Please ensure that the full contributions of each author are acknowledged in the "Add/Edit/Remove Authors" section of our submission form.

**Reviewers' comments:**

Reviewer's Responses to Questions

**Comments to the Authors:**

**Please note that one of the reviews is uploaded as an attachment.**

Reviewer #1: “In Maynard Smith revisited: A multi-agent reinforcement learning approach to the coevolution of signalling behaviour”, authors study the problem of emergence of honest signalling using the Philip Sidney signalling game, using methods from multi-agent reinforcement learning (MARL). Extensive simulations have been conducted showing that proactive pro-sociality i.g. resource provision without signalling, often emerges as the optimal strategy. It challenges the established view of honest signalling. The paper also argues for MARL's value in studying evolutionary dynamics due to its interpretability and ability to study emergent behaviour without pre-defined strategies.

The research problem addressed is novel, interesting and crucially important, and is highly relevant to the audience of PLOS Computational Biology. While the Philip Sidney signalling game has been studied using other methods such as evolutionary game theory, the application of MARL has led to novel and interesting observations that are highly relevant to biological contexts. The analysis in the paper is thorough, including extensive appendix with additional results.

There are some minor issues authors might consider for further improvement of their work.

1) While the paper argues that MARL can be used to study emergent behaviour without pre-defined strategies, it seems the strategy space is still pre-defined (e.g. in Tables 5 and 6). It seems the difference between MARL and evolutionary game theory methods is more about that in the latter, all the strategies are included in model and are always analysed, while in the former, it might the the case that not all strategies might be learned via MARL. Please clarify this.

2) The paper identifies proactive pro-sociality as the key emergent behaviour. While this is a reasonable conclusion given the simulations, it would be important to consider alternative explanations. For example, what is the role of kin selection in here?

Also related to this, there are previous stochastic analyses of the Philip Sidney signalling game which show that a diverse equilibirum points are possible, including those involve providing without signaling, see e.g. "Evolutionary stability of honest signaling in finite populations." 2013 IEEE Congress on Evolutionary Computation. IEEE, 2013. and "Evolution of honest signaling by social punishment." Proceedings of the 2014 annual conference on genetic and evolutionary computation. 2014. This stochastic analysis usually leads to a richer set of possible outcomes (see e.g. "Emergence of cooperation and evolutionary stability in finite populations." Nature 428.6983 (2004): 646-650.) It would be useful to compare MARL with this approach.

3) While it’s good to extend the strategy sets for both donor and beneficial strategies in the paper analysis, it raises the questions if it’s useful to study MARL analysis for the original sets of strategies (for example in the two references above) as the baseline? Otherwise, it would be important to justify why this original setup was not adopted for a MARL analysis (which might allow a more direct comparison?).

3) There are still quite a few typos in the paper. A thorough proofread of the paper is needed.

Reviewer #2: Review uploaded as attachment.

**Have the authors made all data and (if applicable) computational code underlying the findings in their manuscript fully available?**

Reviewer #1: Yes

Reviewer #2: Yes

PLOS authors have the option to publish the peer review history of their article (what does this mean?). If published, this will include your full peer review and any attached files.

Reviewer #1: No

Reviewer #2: **Yes: **Martin Smit

**Figure resubmission:**
---

## [Decision Letter · Decision Letter 1]

30 Apr 2025

PCOMPBIOL-D-24-02076R1

Maynard Smith revisited: A multi-agent reinforcement learning approach to the coevolution of signalling behaviour

PLOS Computational Biology

Dear Dr. Macmillan-Scott,

Thank you for submitting your manuscript to PLOS Computational Biology. After careful consideration, we feel that it has merit but does not fully meet PLOS Computational Biology's publication criteria as it currently stands. Therefore, we invite you to submit a revised version of the manuscript that addresses the points raised during the review process.

Please submit your revised manuscript within 30 days Jun 30 2025 11:59PM. If you will need more time than this to complete your revisions, please reply to this message or contact the journal office at ploscompbiol@plos.org. Please include the following items when submitting your revised manuscript:

We look forward to receiving your revised manuscript.

Kind regards,

Feng Fu

Academic Editor

PLOS Computational Biology

Tobias Bollenbach

Section Editor

PLOS Computational Biology

**Additional Editor Comments :**

One reviewer still has reservations about the revised manuscript. Please (1) Update and document the GitHub code so it is up-to-date and runs out-of-the-box; (2) Provide a detailed, explicit account of the Q-learning implementation, including state definition, discount factor, strategy-space limits -- and demonstrate results for γ = 0 -- to rule out potential artifacts.

**Journal Requirements:**

1) The file inventory includes files for Figures 5a, 5b,6a and 6b. We would recommend either combining these into single Figure 5.tiff and Figure 6.tiff files with separate internal panels, or renumbering them as individual figures, as we are not able to publish multiple components of a single figure as separate files.

**Reviewers' comments:**

Reviewer's Responses to Questions

Reviewer #1: The authors have addressed all my comments very well. I really appreciate their great efforts to make everything much clearer. I believe this is a very good contribution to the theoretical literature of evolution of prosocial and collective behaviours. I am happy to recommend the paper publication in its present form

Reviewer #2: Thank you for addressing the points from my previous review. Firstly, I blatantly misunderstood the "conditional strategies" aspect of the paper and the added sentences about this are much appreciated. I understand how frustrating it can be for a reviewer to completely miss a key aspect of a paper, so thank you for your patience.

That said, I still have a number of concerns with the model used in this paper. Besides the "conditional strategies" misunderstanding, the other overarching point of my first review was about the use of reinforcement learning (RL) in the first place. I thank the authors for clarifying the biological plausibility of RL, I see that, if the focus is individual learning, then it makes sense to model individuals as learning through RL. However, I would like to push back on two things:

1) I am still not happy with the "RL doesn't constrain the strategies that can be learned" argument.

2) I don't understand how the agents can effectively use the discount factor as I don't see how they can predict the next state given their current state and action.

Regarding 1), put bluntly: I disagree. As soon as the state space is determined, the set of learnable strategies is determined. I'm not just being pedantic about choice of words here. Throughout the introduction of the paper (lines 37 to 114), the authors reiterate that, while previous work studying the PS game exists, in this paper

MARL is used so you can study the emergence of decision-making strategies without specifying which ones are learnable. I just don't think this is true. Once you specify the agents' memory length, both agents are learning policies from a now specified strategy space. Please explain whether I am misunderstanding something or justify your use of this statement beyond the explanations given.

Regarding 2), please explain what the next state is. In line 293 you specify that the state in the state-action pair is "made up of both players' actions in the previous round", hence in equation (2) Q(s', a') refers to s' which is the next state, made up of both players' actions in the current round. How can the signaller know their interaction partner's action in the current round? If they don't know what the next action is e.g. if s' is just s (which is the case in the version of the code in the linked Github repository), then this term is just noise and it makes sense that "for each value of learning rate, a lower discount rate leads to the optimal strategy being learned a higher proportion of the time." (line 363). My worry is that the cyclical/alternating strategies are not an emergent property but just a byproduct of the agents thinking they can predict the future i.e. what action their opponent will take, and the various sources of noise in the reward signal. What happens when the discount factor is zero? Are cyclical strategies learned?

I wish I could be more specific in my critiques, but I don't think I understand precisely how Q-learning was used in this paper, and I don't want to critique something I might not understand properly. I tried looking into the code for answers, but I don't think the code is up to date (the last commit was 2 years ago) so I don't want to assume that things were done in a certain way. Please update the code.

**Have the authors made all data and (if applicable) computational code underlying the findings in their manuscript fully available?**

Reviewer #1: Yes

Reviewer #2: **No: **Code is, as far as I can tell, not up-to-date with the current version of the paper.

PLOS authors have the option to publish the peer review history of their article (what does this mean?). If published, this will include your full peer review and any attached files.

Reviewer #1: No

Reviewer #2: No

**Figure resubmission:**
---

## [Editor Report · Decision Letter 2]

8 Jul 2025

Dear Miss Macmillan-Scott,

We are pleased to inform you that your manuscript 'Maynard Smith revisited: A multi-agent reinforcement learning approach to the coevolution of signalling behaviour' has been provisionally accepted for publication in PLOS Computational Biology.

Best regards,

Feng Fu

Academic Editor

PLOS Computational Biology

Tobias Bollenbach

Section Editor

PLOS Computational Biology

---

## [Editor Report · Acceptance letter]

PCOMPBIOL-D-24-02076R2

Maynard Smith revisited: A multi-agent reinforcement learning approach to the coevolution of signalling behaviour

Dear Dr Macmillan-Scott,

I am pleased to inform you that your manuscript has been formally accepted for publication in PLOS Computational Biology. Your manuscript is now with our production department and you will be notified of the publication date in due course.

With kind regards,

Lilla Horvath
